# Can Lateralization of Reverse Shoulder Arthroplasty Improve Active External Rotation in Patients with Preoperative Fatty Infiltration of the Infraspinatus and Teres Minor?

**DOI:** 10.3390/jcm10184130

**Published:** 2021-09-13

**Authors:** Marko Nabergoj, Shinzo Onishi, Alexandre Lädermann, Houssam Kalache, Rihard Trebše, Hugo Bothorel, Philippe Collin

**Affiliations:** 1Valdoltra Orthopaedic Hospital, 6280 Ankaran, Slovenia; mmarkoj@gmail.com (M.N.); rihard.trebse@ob-valdoltra.si (R.T.); 2Faculty of Medicine, University of Ljubljana, Vrazov trg 2, 1000 Ljubljana, Slovenia; 3Department of Orthopaedic Surgery, Faculty of Medicine, University of Tsukuba, Tsukuba 305-8575, Japan; onishishinzo@gmail.com; 4Division of Orthopaedics and Trauma Surgery, La Tour Hospital, 1217 Meyrin, Switzerland; 5Faculty of Medicine, University of Geneva, 1211 Geneva, Switzerland; 6Division of Orthopaedics and Trauma Surgery, Department of Surgery, Geneva University Hospitals, 1205 Geneva, Switzerland; 7Hôpital Saint-Camille, 2 Rue des Pères Camilliens, 94360 Bry-sur-Marne, France; kalache.houssam@gmail.com; 8Research Department, La Tour Hospital, 1217 Meyrin, Switzerland; hugo.bothorel@latour.ch; 9Clinique Victor Hugo 5 Bis Rue du Dôme, 75116 Paris, France; docphcollin@gmail.com

**Keywords:** prosthesis, design, range of motion, degeneration, PROMs, results, complication

## Abstract

(1) Background: Postoperative recovery of external rotation after reverse shoulder arthroplasty (RSA) has been reported despite nonfunctional external rotator muscles. Thus, this study aimed to clinically determine the ideal prosthetic design allowing external rotation recovery in such a cohort. (2) Methods: A monocentric comparative study was retrospectively performed on patients who had primary RSA between June 2013 and February 2018 with a significant preoperative fatty infiltration of the infraspinatus and teres minor. Two groups were formed with patients with a lateral humerus/lateral glenoid 145° onlay RSA—the onlay group (OG), and a medial humerus/lateral glenoid 155° inlay RSA—the inlay group (IG). Patients were matched 1:1 by age, gender, indication, preoperative range of motion (ROM), and Constant score. The ROM and Constant scores were assessed preoperatively and at a minimum follow-up of two years. (3) Results: Forty-seven patients have been included (23 in OG and 24 in IG). Postoperative external rotation increased significantly in the OG only (*p* = 0.049), and its postoperative value was significantly greater than that of the IG by 11.1° (*p* = 0.028). (4) Conclusion: The use of a lateralized humeral stem with a low neck-shaft angle resulted in significantly improved external rotation compared to a medialized humeral 155° stem, even in cases of severe fatty infiltration of the infraspinatus and teres minor. Humeral lateralization and a low neck-shaft angle should be favored when planning an RSA in a patient without a functional posterior rotator cuff.

## 1. Introduction

The treatment of rotator cuff tears was revolutionized with the introduction of RSA, which provides significant improvements in functional and clinical outcomes for many different shoulder pathologies [1]. Studies reporting on long-term outcomes of Grammont-style designs have reported consistent limited restoration of external rotation [2]. This could be explained by the slackening of the remaining rotator cuff or various impingements, since the original Grammont-type RSA design has a medialized center of rotation compared to the native glenohumeral joint [3]. Several biomechanical [4,5] and clinical studies [6] have observed an increase in lateralization which led to improved rotational movements. Thus, the implant design was evolved so that the center of rotation was lateralized compared to the Grammont-type RSA, though remaining medialized compared to the native shoulder joint [2].

Lateralization can be achieved on the glenoid side, the humeral side, or both. It can be promoted by using an additional metal or bone stock on the glenoid side [4], or by using a neck-shaft angle of 135° or 145° as well as a curved or onlay stem on the humeral side [7]. Comparative clinical studies between lateralized and medialized humeral components have been previously reported [8]. However, there are no published studies in the literature that have specifically analyzed the clinical results of primary RSA using a medialized or lateralized humeral component in patients with a nonfunctional posterior rotator cuff.

The purpose of this study was thus to compare ROM and clinical outcomes between different RSA humeral designs in patients with preoperative grade 3 to 4 fatty infiltration of the posterior rotator cuff. The hypothesis was that lateralized RSA using an onlay 145° stem would be associated with an improved external rotation compared to medialized RSA using an inlay 155° stem.

## 2. Materials and Methods

### 2.1. Patient Selection

Between June 2013 and February 2018, 651 RSAs (primary RSA, revision of RSA, and conversion from anatomical shoulder prothesis to RSA) performed by the senior author (P.C.) were considered potentially eligible for inclusion in this retrospective, comparative study using a prospectively collected database. Inclusion criteria consisted of (1) patients who underwent implantation of a primary RSA for rotator cuff arthropathy due to massive rotator cuff tear type E (supraspinatus, infraspinatus, and teres minor) [9], (2) a preoperative grade 3 or 4 fatty infiltration of infraspinatus and teres minor based on the Goutallier classification [10] characterized using non-contrast computer tomography (CT) scans, (3) positive external rotation LAG sign of more than 40° [11], and (4) a minimum follow-up of two years. The exclusion criteria were: incomplete documentation, revision cases, other indication for surgery, and a shorter follow-up.

The included patients were categorized into two groups based on the type of prostheses they received: lateralized RSA (Onlay Group, OG): onlay 145° curved, short stem (lateralized humerus and glenoid); or medialized RSA (Inlay Group, IG): inlay 155° straight standard stem (medialized humerus/lateralized glenoid. Patients were matched in the largest possible ratio (1:1) by age, gender, indication, preoperative range of motion, and Constant score [12].

The study protocol was approved by the hospital ethics committee (CERC-VS-2018-06-1), and all patients gave informed written consent.

### 2.2. Surgical Technique and Implant Design

Patients were operated on under the combination of general anesthesia and interscalene block, and exclusively by a standard deltopectoral approach. An onlay curved short stem with a neck shaft angle of 145° was used in the OG (Ascend Flex, Wright Medical, Memphis, TN, USA), and an inlay straight standard stem with a neck shaft angle of 155° was implanted in the IG (Aequalis II; Wright Medical, Memphis, TN, USA). The stems were impacted with a retroversion of 20°. A bony cylindrical autograft of 7 mm thick was harvested from the native humeral head and systematically used on the glenoid side. The glenoid implant was composed of a 25 mm long peg to safely fix the graft beneath the baseplate, two compression screws, and two locking screws. An angle of 10° of inferior tilt was targeted. A glenosphere with a 36 mm diameter was used [13,14]. Table 1 summarizes the differences in lateralization between the two RSA designs that were implanted in our study.

### 2.3. Postoperative Rehabilitation Protocol

Postoperatively, the arm was placed in a sling for four weeks. Our physiotherapy protocol after RSA was based on three goals. The goal during the first four weeks was to recover the passive anterior forward flexion and external rotation according to a previously validated protocol [3]. After four weeks, the goal was to recover ROM, based on the deltoid reactivation and strengthening in “zero position” according to Saha [15]. The third goal was to recover functional shoulder movements for the daily activities, using neuromuscular techniques to pass from active elevation to functional movements. Strengthening was not recommended.

### 2.4. Study Variables

The main outcomes of interest were the improvements in active external rotation, and in clinical scores in relation to the prosthetic designs. The following patient characteristics were assessed: age, sex, length of follow-up, and ROM.

### 2.5. Clinical Evaluation

All patients were clinically evaluated preoperatively and at the final follow-up. A goniometer was used to assess anterior forward flexion and external rotation for the active ROM assessment. The external rotation was measured with the arm by the side of the body, whereas the internal rotation was measured by the highest vertebral spinous process reached by the patient’s extended thumb. Internal rotation was scored by the following discrete assignment: 0° = 0, buttocks = 1, sacrum = 2, L5 = 3, L4 = 4, L3 = 5, L2 = 6, L1 = 7, Th12 = 8, Th11 = 9, Th10 = 10, Th9 = 11, Th8 = 12, Th7 = 13, Th6 = 14. The assessment included the Constant score [12].

### 2.6. Statistical Analysis

The Shapiro–Wilk test was used to check the normality of distributions. Descriptive statistics were presented in terms of means, standard deviations (SD), medians, and ranges. The significance of pre- vs. postoperative differences within each group was determined using the Wilcoxon signed-rank test for non-normally distributed data and using the paired Student *t*-test for normally distributed data. The significance of differences between groups was determined using the Mann–Whitney U test (Wilcoxon rank-sum test) for non-normally distributed quantitative data, the Student unpaired *t*-test for normally distributed data, and the Fisher exact test for categorical data. Statistical analyses were performed using R version 3.6.2 (R Foundation for Statistical Computing, Vienna, Austria). *p* values < 0.05 were considered statistically significant.

## 3. Results

Forty-seven patients participated in a matched analysis (23 in OG and 24 in IG). Cohorts were comparable in terms of age, gender, surgical indication, preoperative ROM, and Constant score (Table 2).

The postoperative results are summarized in Table 2. Patients in the IG had a significantly greater follow-up compared to the OG (52.0 ± 14.6 vs. 27.3 ± 2.9 months; *p* < 0.001). Anterior forward flexion improved significantly in both groups but was significantly greater postoperatively in the IG compared to the OG (140.4 ± 33.1 vs. 128.9 ± 26.8; *p* = 0.032). External rotation improved significantly only in the OG (preop: 4.6° ± 8.9° vs. postop: 12.0° ± 15.8°; *p* = 0.049) and was also significantly greater postoperatively in the OG compared to the IG (12.0 ± 15.8 vs. 1.9 ± 3.8; *p* = 0.028). In the OG, external rotation improved in 9 cases (10° to 40°), remained comparable in 11 cases, and worsened in 3 cases (5° to 30°). In the IG, external rotation improved in 5 cases (5° to 10°), remained comparable in 17 cases, and worsened in 2 cases (10°). Postoperative internal rotation did not increase in any members of the two groups and was not significantly different between the groups. The Constant score improved significantly in both groups.

## 4. Discussion

The results of this study confirmed our hypothesis; prosthetic designs play a significant role in postoperative active ROM, despite nonfunctional rotator cuffs. Even if functional scores were similar between the two groups, IG had better postoperative anterior forward flexion, and OG a better postoperative active external rotation, even if the infraspinatus and teres minor presented severe fatty infiltration.

We observed a statistically significant increase in external rotation by 7.4° in the OG with lateralized humerus compared to IG with medialized humerus. This result might be related to (1) an increased humeral lateralization, either due to the use of an onlay design or due to the use of a more varus neck-shaft angle stem (145° vs. 155°) [14,16], (2) a tenodesis effect and a retensioning of the remnant posterior cuff (Figure 1), (3) a better recruitment of the posterior deltoid (Figure 1), and (4) less scapular notching [17].

Increased lateralization on the humeral side might have important biomechanical consequences and affect clinical outcomes. This is theorized to increase the tension of the rotator cuff muscles, so that their rotational capacities improve [14,18,19]. Lädermann et al. have shown that the greatest lengthening of the infraspinatus is achieved when a combination of bone increased offset RSA with a 145° onlay stem is used [14]. Several biomechanical studies showed improvement of rotator cuff (especially infraspinatus and teres minor) and posterior deltoid moment arms in lateralized humeral designs [20,21]. The increase in lateralization could potentially improve the length–tension relationship of the posterior remnant of the rotator cuff and thus increase its efficiency. However, the increase of external rotation, we found, may be mainly due to a so-called “tenodesis effect” of the remnant posterior cuff, which could prevent some loss of active external rotation.

Humeral lateralization improves deltoid muscle efficiency. The increase of external rotation could perhaps be explained through the “wrapping effect” of the posterior deltoid when a prosthetic design of lateralized humerus is used [22]. By lateralizing the center of rotation, a major part of the posterior deltoid fibers is preserved for rotational motion, which allows for a possible increase in active external rotation [20,23,24]. The moment arm for the posterior part of the deltoid is approximately 20% of that for the infraspinatus and teres minor [20,23]. Collin et al. showed that patients with an absence of posterosuperior rotator cuff (type E rotator cuff tear) still have an external rotation of 20° at 90° of abduction, potentially generated by the posterior deltoid [9].

A low neck-shaft angle, limiting inferior friction-type impingement, and consequently, scapular notching, could also explain the difference in external rotation between the two groups of the present study [17,25,26,27]. Only one clinical study, performed by Merolla et al., compared the same groups as ours using an OG and IG RSA design with a minimum follow-up of 2 years [28]. Both implants showed similar postoperative ROM between the low (OG) and high (IG) neck-shaft angles, although the former was associated with significantly greater delta scores of external rotation and lower rates of scapular notching [28]. Lateralization seems to play a significant role in scapular notching [29].

Simovitch et al. reported on the minimal clinically important difference (MCID) for different shoulder outcome metrics and ROM after shoulder arthroplasty. They noted that the MCID in terms of active forward anterior forward flexion is 12° ± 4° and for active external rotation is 3° ± 2° [30]. With that knowledge in mind, we can explain why we were not able to find any statistical difference between the clinical outcomes of the OG and the IG. We noticed a significant improvement of movement in one plane in each group. In OG, it was external rotation, whereas in IG, it was abduction, which in the end negated each other; consequently, we could not find a substantial difference between the clinical outcomes of the two analyzed groups.

All previously mentioned findings are crucial when planning RSA in a patient with a loss of active external rotation. Effectively, it has traditionally been implied for this condition that a latissimus dorsi transfer +/− teres major tendon transfer(s) be undertaken [31,32]. Our study demonstrated that an adequate prosthetic design could be sufficient to restore active external rotation, confirming other reports [33]. Consequently, due to the additional difficulty, increased operative time, associated loss of internal rotation [34], and increased neurological complication rate [35], primary transfers do no longer seem justified, as a simple change in prosthetic design could achieve similar results.

### Strengths and Limitations

This study compared two groups of patients that were operated on by the same surgeon, using the same surgical technique, with the same glenoid configuration. Furthermore, the control group was matched according to age, gender, indication, preoperative ROM, and Constant score. This is the first report to specifically analyze the effect of the lateralized humeral stem in primary lateralized glenoid RSA in patients with preoperative third- or fourth-grade fatty infiltration of the infraspinatus and teres minor. We acknowledge, however, several limitations. First, the retrospective design of this study; however, observation and recollection biases were reduced by prospective collection of the data. Second, this is not a randomized study, which might create a sample bias. Third, we did not perform an a priori sample size calculation. Due to a limited number of patients, we did not divide patients within the OG between those who had a satisfying postoperative external rotation and those who did not, preventing analysis of the main predictive factor for this outcome. Lastly, patients in the IG had a significantly longer follow-up compared to those in the OG. As external rotation improves with time [36], the difference in range of motion could have been even more important with a similar follow-up.

## 5. Conclusions

The use of lateralized RSA with a low neck-shaft angle humeral stem results in significantly improved external rotation compared to medialized RSA with a 155° humeral stem, even in cases of severe fatty infiltration of the infraspinatus and the teres minor. Humeral lateralization and a low neck-shaft angle should be favored when planning an RSA in patients without a functional posterior rotator cuff. On the other hand, the medialized humerus with a 155° inlay stem contributed to a greater anterior forward flexion than the other configuration. However, the change in ROM amongst groups did not affect the postoperative clinical outcome.

## Figures and Tables

**Figure 1 jcm-10-04130-f001:**
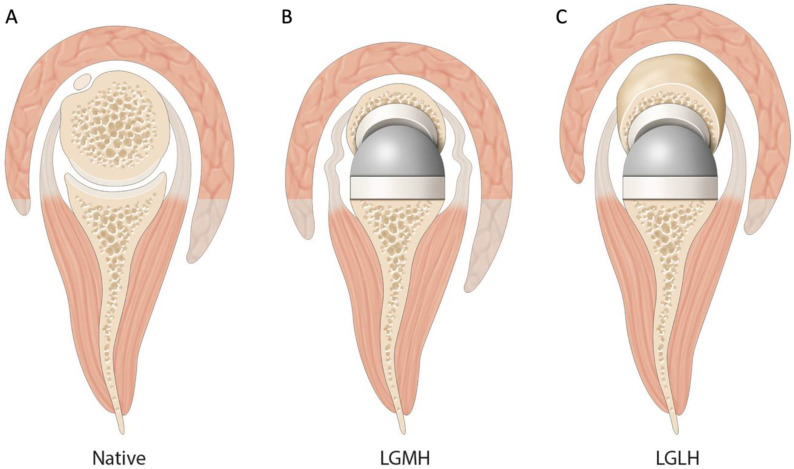
Factors influencing postoperative external rotation. (**A**) Native shoulder. The center of rotation is in the humeral head, and the level of deltoid arm does not allow deltoid recruitment. (**B**) A combination of lateral glenoid/medial humerus RSA. As in native shoulders, the bony lateralization of the center of rotation decreases recruitment of the deltoid for rotation. Additionally, due to the medialized center of rotation compared to the native shoulder, the rotator cuff is slackened and thus less efficient in rotatory motion. (**C**) A combination of lateral glenoid/lateral humerus RSA. Additional lateralization on the humeral side allows important deltoid recruitment and a tenodesis effect and a retensioning of the remnant posterior cuff.

**Table 1 jcm-10-04130-t001:** Lateralization (Expressed in MM) of Different Components Used in Our Study.

Manufacturer	Implant	Gleno-Humeral Construct	Humeral Offset	Glenoid Offset	Global Offset	Glenoid Contribution	Humeral Contribution
Wright	Ascend Flex 145°	LGLH	14.2	17.3	31.5	42%	58%
Wright	Aequalis II 155°	LGMH	8	14.6	22.6	57%	43%

LG—lateralized glenoid, MH—medialized humerus, LH—lateralized humerus, °—degrees

**Table 2 jcm-10-04130-t002:** Comparison Analysis of Pre- and Postoperative Data between Onlay and Inlay Groups.

	Onlay Group (OG, *n* = 23 Patients)			Inlay Group (IG, *n* = 24 Patients)			*p*-Value
	N (%)			N (%)			
	Mean ± SD	Median	(Range)	Mean ± SD	Median	(Range)	
Male sex	9 (39.1%)			9 (37.5%)			1.000
Age at index operation (yrs)	74.6 ± 7.6	77.0	(59.0–87.0)	75.0 ± 5.4	75.0	(65.0–87.0)	0.848
Follow-up (months)	27.3 ± 2.9	28.0	(24.0–31.0)	52.0 ± 14.6	48.5	(24.0–89.0)	<0.001
Anterior forward flexion							
preoperative	92.4 ± 40.3	90.0	(15.0–160.0)	87.9 ± 43.9	80.0	(10.0–165.0)	0.416
postoperative	128.9 ± 26.8	140.0	(70.0–160.0)	140.4 ± 33.1	150.0	(35.0–180.0)	0.032
improvement	36.5 ± 41.8	30.0	(−20.0–145.0)	52.5 ± 41.1	65.0	(−15.0–120.0)	0.112
*p*-value *	<0.001			<0.001			
Internal rotation (°)							
preoperative	4.2 ± 3.3	4.0	(1.0–12.5)	5.1 ± 4.3	4.0	(1.0–13.0)	0.728
postoperative	4.7 ± 2.8	4.0	(1.0–8.0)	3.5 ± 2.7	4.0	(1.0–13.0)	0.109
improvement	0.4 ± 3.7	0.0	(−4.5–7.0)	−1.6 ± 5.1	0.0	(−11.0–6.0)	0.341
*p*-value *	0.726			0.247			
External rotation (°)							
preoperative	4.6 ± 8.9	0.0	(0.0–30.0)	0.8 ± 4.1	0.0	(−10.0–10.0)	0.133
postoperative	12.0 ± 15.8	0.0	(0.0–45.0)	1.9 ± 3.8	0.0	(0.0–10.0)	0.028
improvement	7.4 ± 16.2	0.0	(−30.0–40.0)	1.0 ± 5.1	0.0	(−10.0–10.0)	0.191
*p*-value *	0.049			0.416			
Constant score							
preoperative	33.4 ± 15.5	28.0	(13.0–69.0)	33.3 ± 14.6	32.5	(5.0–65.0)	0.790
postoperative	67.5 ± 14.3	71.0	(35.0–93.0)	67.7 ± 14.1	72.0	(31.0–87.0)	0.898
improvement	34.1 ± 20.4	34.0	(−20.0–80.0)	34.4 ± 19.2	34.5	(3.0–76.0)	0.882
*p*-value *	<0.001			<0.001			

* Between pre- and post-operative measurements. Underlined *p*-values indicate those below 0.05. °—degrees.

## Data Availability

All the available data have been presented in this study. Details regarding where data supporting reported results can be requested at the following e-mail address: hugo.bothorel@latour.ch.

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
