# Peer review of "Can Lateralization of Reverse Shoulder Arthroplasty Improve Active External Rotation in Patients with Preoperative Fatty Infiltration of the Infraspinatus and Teres Minor?"

_jcm, 2021, doi:10.3390/jcm10184130_

Round 1

Reviewer 1 Report

The paper is overall well written and cohorts are matched well.  Frankle has shown this already with a lateralized system but you fail to make reference to this article and I would suggest including in discussion.  Overall well written and acceptable for publication 

Author Response

Reviewer #1's comments:

Comments

Answers/Corrections

Line

The paper is overall well written and cohorts are matched well.  

Thank you for your comment. Please see our answers below.

Frankle has shown this already with a lateralized system but you fail to make reference to this article and I would suggest including in discussion.  Overall well written and acceptable for publication 

Added. We were aware of the reports by Frankle et al. (The Reverse Shoulder Prosthesis for glenohumeral arthritis associated with severe rotator cuff deficiency. A minimum two-year follow-up study of sixty patients). However, Frankle and al. achieved lateralization on the glenoid side and not on the humeral side as it was done in our study. The reference has been consequently added in the Introduction and not in the Discussion.

44, 45, 50, 52, 79, 80, 82, 91, 103, 112, 126, 167, 169, 179, 181, 183,, 190, 192, 193, 195, 198, 199, 202, 206, 213, 215, 216, 231, 266-424.

Reviewer 2 Report

Materials and Methods 

lines 59-72 should be removed

lines 84-87 how patient's were included in one or other group?randomized? any clinical or image criteria?

47 implants in 5 years is less than 1 implant a month, how many implants in total were perfomed in that time period?

Statystical analysis

any power analysis to justify sample size?

difference in the results between groups should be analyzed

Results:

comparison and differences of the results between groups should be reported ( is there a significant difference in AFF IR or ER between the groups?). I see significant difference between pre and postop within groups but not BETWEEN groups

table 2:

- difficult to read, should be re-edited allowing easier consultation

- in the ER area, the underlined value i think is the wrong one 

Discussion:

lines 151-153 should be removed

you should discuss difference of the results between groups and not only post-op and pre-op gf each group (i.e AFF)

Author Response

Reviewer #2's comments:

Comments

Answers/Corrections

Line

Materials and Methods 

Lines 59-72 should be removed

Removed.

60-72

Lines 84-87 how patient's were included in one or other group?randomized? any clinical or image criteria?

Added. All patients having a lateralized RSA (onlay 145° curved short stem), a medialized RSA (Inlay 155° straight standard stem) and matched our inclusion criteria were found. Then we matched them in the largest possible ratio (1:1) by age, gender, indication, preoperative range of motion and Constant score.

74-91

47 implants in 5 years is less than 1 implant a month, how many implants in total were perfomed in that time period?

656 RSA were performed in that time period. This information has been added. E-type lesions are rare, representing only 15% of massive rotator cuff (Collin, Relationship between massive chronic rotator cuff tear pattern and loss of active shoulder range of motion, JSES 2014).

74, 75.

Statistical analysis

Any power analysis to justify sample size?

We did not perform an a-priori sample size calculation. We added that under the limitations.

226, 227

Difference in the results between groups should be analyzed

As written in Statistical analysis – “The significance of differences between groups was determined using the Mann-Whitney U test (Wilcoxon rank-sum test) for non-normally distributed quantitative data, the Student unpaired t-test for normally distributed, and the Fisher exact test for categorical data”, the results of each group have been compared between the groups. They are reported in Table 2.

143, 145, 146, 147, 151, 152.

Results:

comparison and differences of the results between groups should be reported ( is there a significant difference in AFF IR or ER between the groups?). I see significant difference between pre and postop within groups but not BETWEEN groups

Added. The results of each group have been compared between the groups and they are reported in Table 2. Under the Results section we also added the comparison between the groups for ER and IR, whereas the comparison for AFF is already reported.

145-147, 151.

table 2:

- difficult to read, should be re-edited allowing easier consultation

Re-edited

152

- in the ER area, the underlined value i think is the wrong one 

Corrected.

152

Discussion:

lines 151-153 should be removed

Removed.

151-153

You should discuss difference of the results between groups and not only post-op and pre-op gf each group (i.e AFF)

Hypothesis of our study was that lateralized RSA using an onlay 145° stem would be associated with an improved external rotation compared to medialized RSA using an inlay 155° stem (even in the absence of functional external rotators). Thus, in our discussion, we only focused on the differences between the groups in terms of external rotation (the significant improvement of external rotation in the OG and significantly higher ER in the OG compared to IG) and did not discuss the difference of AFF and IR.

Results section and Table

Round 2

Reviewer 2 Report

thank you for answering my queries